# The Pyramiding of Elite Allelic Genes Related to Grain Number Increases Grain Number per Panicle Using the Recombinant Lines Derived from *Indica–japonica* Cross in Rice

**DOI:** 10.3390/ijms24021653

**Published:** 2023-01-14

**Authors:** Xuhui Liu, Xiaoxiao Deng, Weilong Kong, Tong Sun, Yangsheng Li

**Affiliations:** 1State Key Laboratory of Hybrid Rice, College of Life Sciences, Wuhan University, Wuhan 430072, China; 2Shenzhen Branch, Guangdong Laboratory for Lingnan Modern Agriculture, Genome Analysis Laboratory of the Ministry of Agriculture, Agricultural Genomics Institute at Shenzhen, Chinese Academy of Agricultural Sciences, Shenzhen 518120, China

**Keywords:** *indica–japonica* hybrid, grains number per panicle, elite allelic gene, genotype combinations

## Abstract

*Indica*(*xian*)-*japonica*(*geng*) hybrid rice has many heterosis traits that can improve rice yield. However, the traditional hybrid technology will struggle to meet future needs for the development of higher-yield rice. Available genomics resources can be used to efficiently understand the gene-trait association trait for rice breeding. Based on the previously constructed high-density genetic map of 272 high-generation recombinant inbred lines (RILs) originating from the cross of Luohui 9 (*indica*, as female) and RPY geng (*japonica*, as male) and high-quality genomes of parents, here, we further explore the genetic basis for an important complex trait: possible causes of grain number per panicle (GNPP). A total of 20 genes related to grains number per panicle (GNPP) with the differences of protein amino acid between LH9 and RPY were used to analyze genotype combinations, and PCA results showed a combination of *PLY1, LAX1, DTH8* and *OSH1* from the RPY geng with *PYL4, SP1, DST* and *GNP1* from Luohui 9 increases GNPP. In addition, we also found that the combination of *LAX1-T2* and *GNP1-T3* had the most significant increase in GNPP. Notably, Molecular Breeding Knowledgebase (MBK) showed a few aggregated rice cultivars, *LAX1-T2* and *GNP1-T3*, which may be a result of the natural geographic isolation between the two gene haplotypes. Therefore, we speculate that the pyramiding of *japonica*-type *LAX-T2* with *indica*-type *GNP1-T3* via hybridization can significantly improve rice yield by increasing GNPP.

## 1. Introduction

Rice (*Oryza sativa* L.) is one of the most important food crops, and more than half of the world’s population relies on rice as a staple food [1]. Especially in Southeast Asia, rice production directly affects social stability and national security [2]. However, the global rice production is falling far short of worldwide rice demand. Because of reductions in available land and the increasing global population, it has become particularly urgent to increase rice production [2]. This is the reason for the modern molecular breeding of rice based on diverse germplasm resources and genetic background, gathering known genes to improve rice breed [3,4,5]. It is well known that rice yield is mainly composed of four indicators: grains number per panicle (GNPP), seed setting rate (SSR), effective panicle number (EPN) and thousand grain weight (TGW). However, the crop yield as a typical quantitative trait is regulated by numerous genes. In addition, environmental factors can even overshadow the effect of genes, hindering research [6]. Nevertheless, about 2296 genes have been successively identified in rice in the past, of which 189 genes related to rice grain yield have been cloned and functionally validated [7]. These genes regulate the development and morphology of rice panicles, ultimately affecting rice yield [8]. Notably, there is a contradiction among the four indicators of rice yield hindering rice breeding; for example, increasing the grains number per panicle will lead to a decrease in the effective panicle number [9]. So, simply aggregating yield-enhancing genes may not yield the desired result [10]. However, in previous studies, we found that high-yielding rice varieties often have higher grain number per panicle, implying a significant positive correlation between grain number per panicle and yield. It is feasible to increase rice production by increasing the grain number per panicle, but the mechanisms underlying the interrelationships of GNPP-related genes are still not fully clarity [11].

Rice hybrids, especially *indica–japonica* hybrids, have shown a significant hybrid advantage, which are undoubtedly suitable materials for studying the mechanism of genotype combinations [12]. Previous works in our laboratory found that the progeny of the cross between *indica* LuoHui 9 (LH9, female) and *japonica* RPY geng (RPY, male) had an obvious hybrid advantage and showed a significant increase in GNPP and yield [13]. Unfortunately, as F1 is a temporary genetic population, the heterozygous F1 rice is difficult to preserve and limit due to the heterozygosity of the genome, which makes it difficult to identify gene combinations [4]. Therefore, in the present study, recombinant inbred lines (RILs) from LH9 and RPY were used to explore the gene combinations for high GNPP, and we observed that some lines in the RILs showing more suitable traits for agricultural production. Thus, we speculate that some genetic combination of the parents causes this phenomenon, which may be used in rice breeding and agricultural production.

The development and maturation of second-generation sequencing technologies lead to a decreased cost of resequencing. The generated genomic resources, resequencing of various diverse germplasm lines, would aid in identifying and exploring allelic/haplotype variations, thus harnessing genetic diversity [14,15]. Meanwhile, many databases containing rice phenotypes and genetic data have emerged, such as MBK [16], helping us to find out the association between traits and haplotypes and facilitating the utilization of genes regulatory networks. In the present study, we analyzed the superior haplotypes and the haplotype combination of many important genes affecting the GNPP using 272 RILs, which provide an insight into the yield enhancement mechanism of *indica–japonica* hybrid rice.

## 2. Results

### 2.1. GNPP Distribution of RILs Populations Cross from Indica LH9 and Japonica RPY

During the years 2016–2019, the GNPP traits of 272 RILs were relatively stable (Figure 1A), which indicated that most genes in the population existed in pure form. Comparing the GNPP of each year, there was a clear similarity in the RIL population (Figure 1A,B), indicating that there was not a significant difference in the characteristics of the GNPP in each year. Then, using the GNPP as a basis for hierarchical clustering of the RILs population, with similar individual clustering into the same group, we divided the population into three groups, respectively: the Low group (L), the Middle group (M) and the High group (H) (Figure 1C,D). It is generally believed that the *indica–japonica* hybrid rice yield is affected by its parental homology [15]. However, we did not find significant differences between the RIL groups when comparing the blood rate of the *indica* LH9 (Figure 1E), which implied that the genome fragments of the *japonica* RPY mixed between groups were similar. These results indicate that the differences between groups may be due to the differences in the alleles at the genomic level.

### 2.2. Significant Genetic Background Differences between Parents

We genotyped 31 genes related to the GNPP, retaining only those genes in which differed in protein sequence (Table 1). Previous studies reported that these 31 known genes affected the yield by influencing the formation and structure of primary and secondary branching peduncles in rice [3,16]. Based on the recently assembled high-quality genomes of LH9 and RPY, we characterized the sequence differences of these important genes in the parents [17]. Of these, all genes from the *japonica* RPY rice were genotypically identical to *japonica* Nipponbare, except for *IPA1*. Twenty genes showed differential genotypes between LH9 and RPY, with a difference rate of 67%, which contributes to the basis of significant heterozygous advantaging in the progeny of LH9 and RPY crosses. *NOG1, LAX1, LP, An-1, LAX2, DTH7, GAD1, DEP1* and *SP1* were all predicted to have severe amino acids changes that may affect protein function (Table 1). Although the remaining 11 genes also have missense mutations, their protein functions were predicted to be unaffected by amino acid substitutions.

### 2.3. Principal Component Analysis Reveals Superior Genotype Combinations

To determine the role of parental genotypes in GNPP of RILs, we screened and scored the RILs based on the mean of GNPP from 2016 to 2019 (Table 2). Moreover, these RILs were separately distributed using principal component analysis (Figure 2A), indicating that PC1 is an important factor contributing to the difference in GNPP in RILs. It is evident from the loadings that the main contributing genes are *PLY1, LAX1, OSH1, SP1, DTH8, DST, PYL4* and *GNP1* (Figure 2B). So, we speculate that some combination of these genes with specific genotypes may effectively increase the GNPP. The effects of *LAX1* and *GNP1* on GNPP were considered, as they were the largest and smallest loadings of PC1 and PC2, respectively (Figure 2B). Meanwhile, the effects of *LAX1* and *GNP1* on rice panicle development were both positively regulated (Table 1) and have been reported to have significant QTN within gene [18,19,20]. The effect of *LAX1’s* QTN on GNPP was inconsistent with previous findings, while *GNP1* was consistent with previous studies (Figure 2C,D). The CA-QTN combination of *LAX1* and *GNP1* had a significant effect on GNPP (Figure 2E). Yield data of 533 rice materials from Huazhong Agricultural University also support our result (Figure 2F). This suggests that the combination of *LAX1* of *japonica* genotype and *GNP1* of *indica* genotype is expected to increase the number of grains per panicle, and finally increase the yield of rice.

### 2.4. Haplotype Analysis of Target Genes and Their Geographic Origin

The KnownGene MBK database was used for haplotype analysis, only considering SNPs in the gene region (Figure 3A,D). *LAX1-T2* and *GNP1-T1* are the haplotypes of the paternal *japonica* RPY, *LAX1-T4* and *GNP1-T3* are the maternal haplotypes of *indica* rice LH9. There is a clear *indica*/*japonica* tendency between *LAX1* and *GNP1* haplotypes (Figure 3B,E) and *indica* LH9 and *japonica* RPY are typical *indica* and *japonica* haplotypes, respectively. In the MBK data, the GNPP mean of the T4 haplotype of *LAX1* was smaller than that of the other haplotypes (Figure 3C). This also supports our previous view that the C base of *LAX1’s* QTN (chr3-35558484) is superior to the A base, which is inconsistent with previous research results [21]. The haplotype analysis of *GNP1* was as expected, with the T3 haplotype carrying the A base QTN (chr3-36150781) with the highest GNPP mean (Figure 3F). The T2 haplotype of *LAX1* and the T3 haplotype of *GNP1* have great differences in geographical distribution (Figure 3G,H), which explains why there are only six rice cultivars with both *LAX1-T2* and *GNP1-T3* in the MBK database, and the reason for the low percentage among 5000 rice cultivars.

### 2.5. Specific Combinations of Indica–japonica Alleles Increase the GNPP

In the present study, a high-resolution bin-based linkage map was based on sequencing from *indica* LuoHui9 (female), *japonica* RPYgeng (male) and 272 RILs. We analyzed 31 known genes associated with GNPP. Twenty genes with 10 deleterious mutations were predicted by PROVEAN assessing the effects of amino acid alterations (Table 1, Figure 4). The rest of the genes had no protein sequence differences. In previous studies [22], PROVEAN was shown to accurately predict the effect of amino acid changes on protein function. The missense mutation SNPs in the exon were used for genotyping, and it could be found that LH9 and RPY represented typical *indica* and *japonica* genotypes, respectively (Table 3). To identify the key genes for the difference in GNPP, we performed score statistics based on the mean GNPP each year of parental genotypes, and PCA’s result showed that eight genes, *PLY1, LAX1, OSH1, SP1, DTH8, DST, PYL4* and *GNP1*, might be the main factors responsible for the difference in our RILs population. The grouping based on hierarchical clustering showed that the frequency of superior genotypes of candidate genes gradually increased in the Low, Middle, and High groups, confirming that these superior genotypes increased GNPP. As a complement, these genotypes were also verified using the MBK database, with *PLY1*, *LAX1*, *DTH8* and *OSH1* genotypes of RPY exhibiting higher GNPP, while *PYL4*, *SP1*, *DST* and *GNP1* exhibited high GNPP for the genotype of LH9 (Table 3).

## 3. Discussion

Rice, one of the most important food crops, is used as a staple food by more than half of the world’s population. Although the breeding of new varieties with improved traits was hampered in the past by the lack of information on the origin of desirable alleles and the genetic background of donor lines, the genetic background of many rice varieties and their genotypes have now been identified with the development of second-generation sequencing technology and the advancement of the rice genome project [23]. With the advancement of rice research in recent years, many functional genes have been identified and used in rice breeding [24]. The emergence of rice genetic information resource databases has made it tremendously convenient to construct superior rice lines and has made the association between genes and traits increasingly clear [25]. Among these databases, the MBK Rice Resource Database, which contains genomic information of over 5000 rice lines, contains 127 types of traits with a total of over 4.8 million records [16].

Huang et al. proposed the idea of promoting rice breeding by aggregating superi-or genotypes through QTN [23]. As expected, GNP1 had the effect of enhancing GNPP with an A base at Chr3-36150781 QTN. But notably, rice carrying an A base at position Chr1-35558484 on the LAX1 gene had fewer GNPP than those carrying a C base, as verified in our RILs population and mean GNPP of haplotype from MBK database (Figure 2C, Figure 3C). Rice varieties with GNPP over 200 in MBK that LAX1 also corresponds to the T2 genotype, which carries a C base at Chr1-35558484 QTN (Table 4). We found that the T2 genotype of LAX1 may had a greater effect on increasing GNPP than the T4 genotype.

This paper presents an idea that rice GNPP, as a typical quantitative trait, is mainly influenced by the number of genes, but some combinations of superior genotypes also play a crucial role in enhancement. In our *indica–japonica* RILs population, the combination of *PLY1*, *LAX1, DTH8* and *OSH1* from the paternal *japonica* genotype with *PYL4, SP1, DST* and *GNP1* from the maternal *indica* genotype can enhance rice GNPP. Furthermore, PCA revealed that *LAX1-T2* and *GNP1-T3* had the highest and lowest contributions to the loadings of PC1 and PC2, respectively (Figure 2B). *LAX1*, a floral meristem-specific gene encoding a plant-specific bHLH transcription factor, is a major regulator of axillary shoot primordium formation in rice [26]. Excellent allelic genotypes of *GNP1* can increase cytokinin activity in rice spike cells, such as *GNP1-T3* [27]. These genes affect the distribution and activity of phytohormones on rice spike primordia by respective signaling pathways, which in turn regulate the number of primary and secondary branching peduncles.

## 4. Materials and Methods

### 4.1. Plant Materials

In 2016–2019, the 272 RILs and parents were planted in the experimental field of the Ezhou Experimental Base of Wuhan University (30° N, 114° E), Breeding Experimental Base of Wuhan University, Tianyuan Co., Ltd. in Hannan District (30° N, 114° E), Wuhan City, Hubei Province (from mid-May to October) or the Hybrid Rice Experimental Base of Wuhan University in Lingshui City (18° N, 110° E), Hainan Province (from December to April of the next year). All plants were under standard agricultural planting management.

### 4.2. SNP Variation and Effect Prediction

RiceVarMap v2.0 was utilized to check the impact of variations (SNPs and InDels) on gene function [28,29]. The distribution of variants was obtained in terms of intron, exon, splice region, 3 prime and 5 prime UTR variants. The Protein Variation Effect Analyzer (PROVEAN) tool was used to gauge the deleterious effect of SNPs. Prediction of the SNP impact on the biological function was obtained for the amino acid changes and PROVEAN score at threshold −2.5. If SNP scores ≤−2.5, it is predicted to be ‘deleterious’ to the protein function, whereas values > 2.5 predict ‘neutral’ effects of sequence variations [25].

### 4.3. Haplotype Variation in Genes Related to Grains Number per Panicle

For the MBK database, a total of 5280 samples were used to determine the locus genotype (allele), and the genotype with sample number >=10 was summarized [16]. For each locus, the number of genotypes can represent the number of alleles in the population. KnownGene of the MBK database was used to correspond to the genotypes of the parents, based on the SNPs obtained via resequencing.

### 4.4. Identify Superior Haplotypes of the Target Genes

Because of the limitation of genetic diversity in the RILs population, there will be a certain bias in identifying the superior allele, which may lead to a mistake. So, the phenotype data from MBK, the rice germplasm resource database, was used to determine which parent genotype is the superior allele in GNPP trait (Table 3). As expected, the genotypes of LH9 and RPY are *indica* type and *japonica* type, respectively. In addition to the GNPP, we also investigated other yield-related traits (Table 3) and the GNPP showed negative correlation with the EPN.

### 4.5. Hierarchical Clustering and Principal Component Analysis

All the recorded phenotypic data were from five crop seasons. The factoextra R package (https://CRAN.R-project.org/package=factoextra, accessed on 8 April 2021) based on version 3.6.3 of R uses all the GNPP dates of RILs to perform hierarchical clustering, and the Low, Medium and High GNPP groups contain 75, 108 and 89 RILs, respectively. Individual RILs within each group had similar GNPP, while there were significant GNPP differences between each group. For PCA, first, we performed GNPP mean statistics on the 20 GNPP-related genes with parental differences. The parental genotypes were then scored using the mean of GNPP (Table 2), and parental genotype scores were assigned to the RILs population. Finally, principal component analysis of populations of RILs assigned the genotype scores to find out the key genes.

### 4.6. Statistical Analysis and Visualization

T-test analysis was performed to test statistical significance to understand the phenotypic performance of each haplotype. Different alphabets denote significant difference and vice versa. Furthermore, only the haplotypes validated in parents were considered for statistical analysis. The visualization of all statistical charts is based on the R 3.6.3 version, which was first completed by using the ggplot2 R package (https://CRAN.R-project.org/package=ggplot2, accessed on 8 April 2021), and then adjusted and modified.

## 5. Conclusions

The grains number per panicle, seed setting rate, effective panicle number and thousand grain weight are important factors affecting rice yield, and there is a balance among these factors that hinders the improvement of rice yield. The hybrid progeny of *indica* LuoHui9 and *japonica* RPYgeng have obvious heterosis, such as a greater number of grains per panicle, and the improvement in the GNPP can effectively increase the yield of rice. To investigate the mechanism, we developed a high generation (>F15) of 272 RILs derived from LH9 and RPY. Using deep resequencing data, a high-density genetic map containing 4758 bin markers was constructed, with a total map distance of 2356.41 cM. We tracked and recorded the GNPP data of the five plantings of the RILs population and used bioinformatics methods for analysis, and it was found that eight genes influenced the GNPP with genotype differences: *PLY1-T3, LAX1-T2, DTH8-T1* and *OSH1-T3* from the paternal *japonica* genotype and *PYL4-T6, SP1-T2, DST-T2* and *GNP1-T3* from the maternal *indica* genotype. Among them, *LAX1-T2* and *GNP1-T3* had the greatest effect on the GNPP of the RILs population. The germplasm data from the MBK database were compared for further analysis, and we found that the *LAX1* gene differed from the previous studies in affecting rice GNPP, while many materials emerged for high GNPP with the *LAX1-T2* haplotype. Meanwhile, it was found that there is a natural geographical isolation between rice varieties carrying *LAX1-T2* and rice varieties carrying *GNP1-T3*. Among the more than 5000 rice varieties in the MBK database, only 6 varieties have both *LAX1-T2* and *GNP1-T3*, which also verifies the existence of this geographical isolation. Therefore, we speculate that the combination of *LAX1-T2* and *GNP1-T3* using rice hybridization technology can increase the GNPP and ultimately serve the purpose of increasing rice yield. This finding is valuable for improving rice varieties to increase the number of grains per panicle.

## Figures and Tables

**Figure 1 ijms-24-01653-f001:**
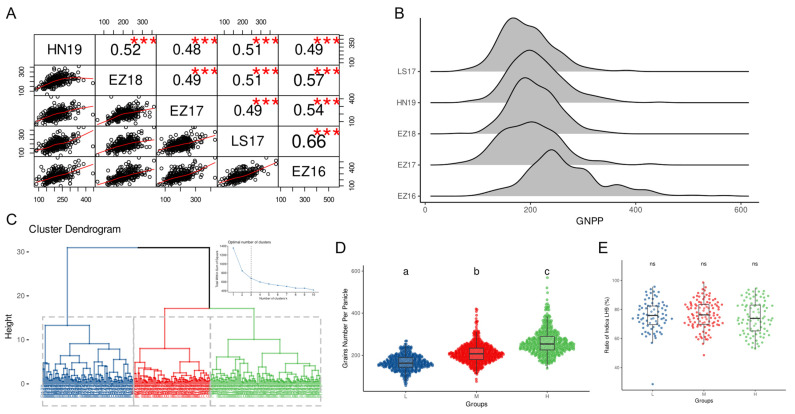
Distribution of Grain Number per Panicle of the RILs population. (**A**) Correlation analysis of the GNPP of rice in five times plantings of the RILs population; (**B**) distribution of GNPP of the RILs population; (**C**) the hierarchical clustering result of RILs population according to distribution of GNPP—blue indicates Low, red indicates Medium and green indicates High; (**D**) differences in the GNPP between L, M and H groups; (**E**) consanguinity rate of female *indica* LH9 rice in L, M and H groups. *** indicates *p* value < 0.001. Different alphabets denote significant difference and vice versa. ns indicates no significant difference.

**Figure 2 ijms-24-01653-f002:**
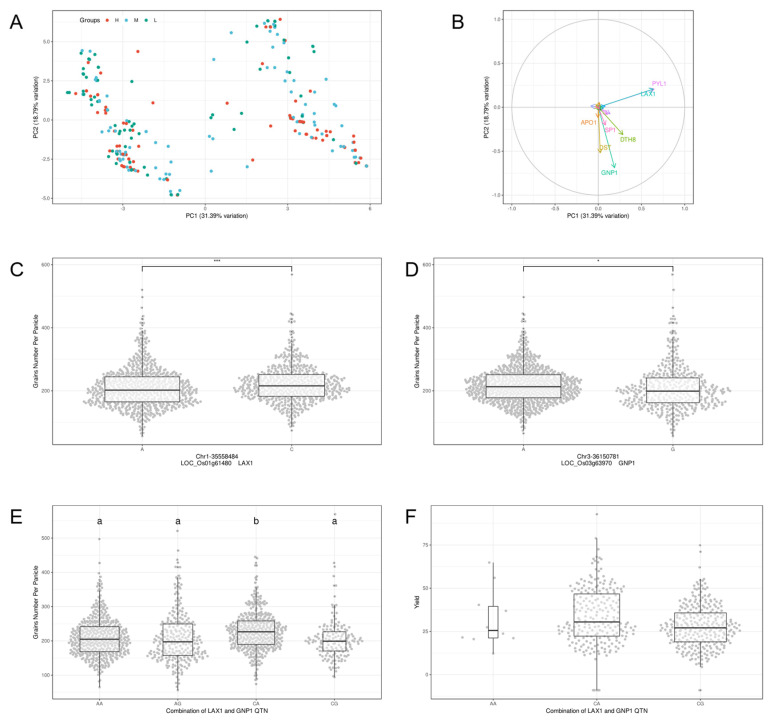
Principal component analysis reveals superior genotype combinations. (**A**) Principal component analysis of 20 GNPP-related genes with parental genotypes differences in RILs population; (**B**) parental differential gene loadings for principal component 1 and principal component 2; (**C**) differences from the QTN of LAX1 for GNPP in the RILs population; (**D**) differences from the QTN of GNP1for GNPP in the RILs population; (**E**) QTN combination of LAX1 and GNP1 affecting GNPP in the RILs population; (**F**) effects of QTN combination with LAX1 and GNP1 on the yield of HAU533 rice materials. * indicates *p* value < 0.05. *** indicates *p* value < 0.001. Different alphabets denote significant difference and vice versa.

**Figure 3 ijms-24-01653-f003:**
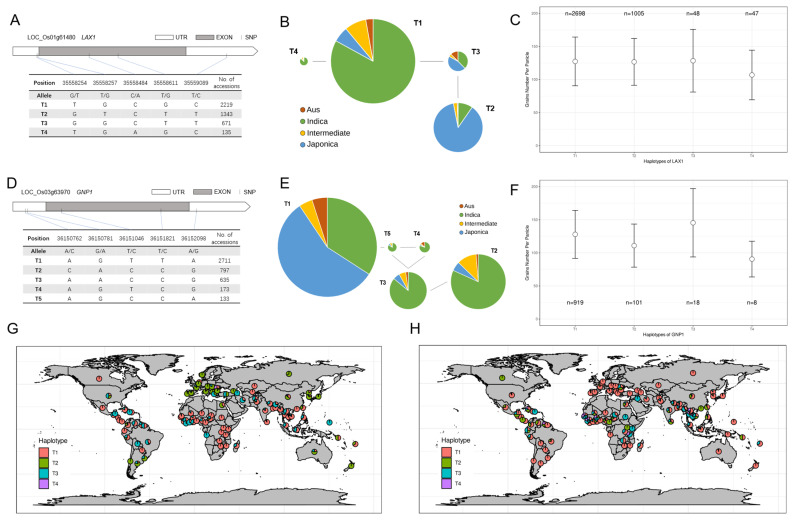
Haplotype analysis of *LAX1* and *GNP1* using the MBK database. The number of haplotypes of the *LAX1* (**A**) and *GNP1* (**D**) gene is based on the SNP positions in the MBK database. The relationship between the LAX1/GNP1 haplotypes (**B**,**E**); (**C**) the difference in GNPP among the *LAX1* (**C**) and *GNP1* (**F**) haplotypes recorded in MBK; n means the number of records. The geographical distribution of various haplotypes of *LAX1* (**G**) and *GNP1* (**H**).

**Figure 4 ijms-24-01653-f004:**
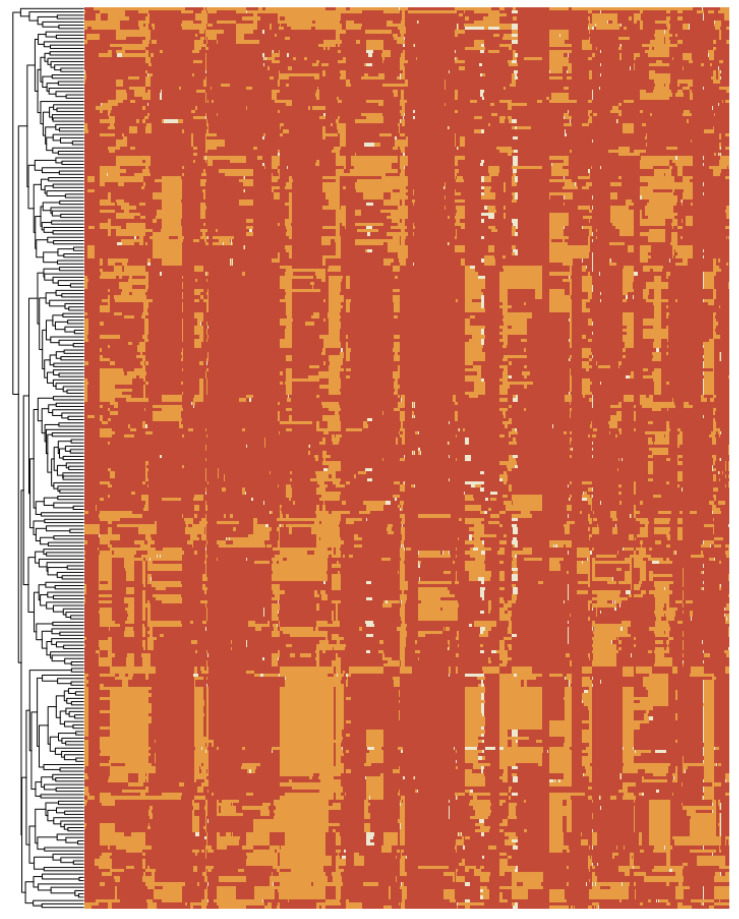
Similarity clustering of 272 RILs population based on 4758 bin markers. Red indicates the male parent, yellow indicates the female parent and white indicates the recombination/NA region.

**Table 1 ijms-24-01653-t001:** Details for differences in the known genes related to the grains number per panicle between LH9 and RPY in rice.

Gene	RGAP Locus ID	GNPP	PRB	SRB	Number of Mutations	Functional Impact of Mutations
*Gn1a*	LOC_Os01g10110	-	-	-	4	N535K, H116R, G54A, A79_A80del
*NOG1*	LOC_Os01g54860	+			1	E346del *
*PYL1*	LOC_Os01g61210	-	-	-	1	F49C
*LAX1*	LOC_Os01g61480	+	+	+	2	D74E *, S117A
*LP*	LOC_Os02g15950	-	-	-	2	L3fs *, S32fs *
*PYL4*	LOC_Os03g18600	-	-	-	1	A86P
*OSH1*	LOC_Os03g51690	+	+	+	1	Q23_H24dup
*DST*	LOC_Os03g57240	-	-	-	2	T201dup, A124_V125insAAAAAV
*GNP1*	LOC_Os03g63970	+		+	1	V41A
*An-1*	LOC_Os04g28280	-	-	-	1	Q87fs *
*LAX2*	LOC_Os04g32510	+		+	8	H65_H66dup, T131_P138del, L177P, A180T, P210A, R225M *, A237del, A237V
*APO1*	LOC_Os06g45460	+	+	+	3	G292_G294del, R204G, I17V
*DTH7*	LOC_Os07g49460	+	+	+	8	D68E
*DTH8*	LOC_Os08g07740	+	+	+	8	N295S
*PAY1*	LOC_Os08g31470	+		+	2	W2R, P150T
*GAD1*	LOC_Os08g37890	+			1	R101fs *
*IPA1*	LOC_Os08g39890	+	+	+	1	L292I
*DEP1*	LOC_Os09g26999	+	+	+	3	L228H, Q283fs *, C324S
*TAW1*	LOC_Os10g33780	+	+	+	1	A33_A34insSASA
*SP1*	LOC_Os11g12740	+	+	+	5	A550_G551del, D475_G476del *, A401G, V328A, H301_A306del

GNPP—grains number per panicle; PRB—primary branch; SRB—secondary branch. +—promote; -—inhibit; *—deleterious mutations.

**Table 2 ijms-24-01653-t002:** Score of parental genotypes in the mean of grains number per panicle in five years.

Gene	Genotype	HN19	EZ18	EZ17	LS17	EZ16	Score
*Gn1a*	F	182.6497	173.4107	167.1413	152.6666	212.2488	0
	M	213.6251	209.3672	201.3538	194.33	266.876	5
*NOG1*	F	216.5889	213.6555	209.3696	191.0993	261.7751	3
	M	209.5623	204.0087	194.7091	192.5892	266.6108	2
*PYL1*	F	214.8368	212.1481	208.8955	197.772	267.9565	5
	M	211.2413	204.1541	192.9816	188.7112	262.2566	0
*LAX1*	F	217.2252	214.2353	210.1405	199.207	270.9662	5
	M	209.5813	202.1089	192.1287	187.8076	261.0606	0
*LP*	F	222.9414	202.7627	200.4978	204.8714	260.9552	2
	M	213.5856	211.2246	202.819	192.9035	268.4735	3
*PYL4*	F	198.6192	185.3971	171.8941	160.8207	219.5586	0
	M	210.525	209.27	200.3168	192.4851	265.109	5
*OSH1*	F	212.5358	222.7958	191.7582	199.8374	289.2342	4
	M	211.9244	203.0429	200.4007	189.2127	255.0527	1
*DST*	F	201.8318	200.084	187.2243	187.9039	234.3329	0
	M	219.2916	211.2264	207.4977	196.3835	275.7076	5
*GNP1*	F	201.8206	199.0928	191.335	192.5	259.8899	0
	M	217.3728	211.268	204.5793	192.7047	266.2327	5
*An-1*	F	209.6562	215.0945	202.7716	188.6553	260.9996	2
	M	213.6572	200.3582	199.252	198.3246	268.3011	3
*LAX2*	F	212.1621	210.6081	200.55	189.7581	251.7476	3
	M	211.6038	206.166	199.4247	193.7761	274.4033	2
*APO1*	F	208.4567	200.882	198.3117	195.7946	246.0994	1
	M	213.6812	208.9489	200.5811	191.586	268.7792	4
*DTH7*	F	216.7544	203.4692	219.8124	200.7108	274.3199	4
	M	211.679	207.5063	198.7814	191.6447	263.6816	1
*DTH8*	F	219.7284	210.7304	206.7825	196.22	271.8351	5
	M	209.1165	206.4734	196.9023	190.8444	262.2673	0
*PAY1*	F	218.0417	214.4899	214.4749	192.9009	278.6728	4
	M	212.2218	207.3872	196.8358	193.6027	263.799	1
*GAD1*	F	213.9167	210.7948	204.68	192.4931	261.6666	3
	M	211.1683	207.3715	196.0528	194.2355	262.2458	2
*DEP1*	F	211.652	220.5395	205.0621	226.2993	266.4615	4
	M	212.3448	206.3899	199.034	188.2502	263.9062	1
*TAW1*	F	212.9674	192.356	187.9528	198.0376	253.4211	2
	M	212.3691	210.8745	203.1138	191.4399	267.4437	3
*SP1*	F	217.9557	219.8128	206.507	199.3304	278.6635	5
	M	209.8068	203.1178	194.7337	188.4645	257.744	0

F—*japonica* RPY gene genotype; M—*indica* Luohui9 genotype.

**Table 3 ijms-24-01653-t003:** Parental genotypes in the MBK database corresponding to genotypes and yield traits.

Gene	ID	Genotype		GNPP	FERT/%	EPN	TGW/g	GYPM/kg	Aus	Indica	Japonica	Intermediate	Total
*PYL1*	LOC_Os01g61210	LH9	T1	127.2 ± 47.22	80.16 ± 9.94	9.02 ± 3.54	25.1 ± 3.87	396.14 ± 102.47	52	1779	103	56	1990
		RPY	T3	131.42 ± 33.65	81.13 ± 9.98	10.45 ± 2.73	26.3 ± 4.88	518.73 ± 105.69	12	98	751	75	936
*LAX1*	LOC_Os01g61480	LH9	T4	112.06 ± 39.1	74.87 ± 16.54		26.06 ± 3.42	367.73 ± 71.71	0	134	4	0	138
		RPY	T2	128.06 ± 35.34	81.22 ± 9,42	9.56 ± 2.82	25.48 ± 2.53	549.23 ± 90.84	0	65	1304	23	1392
*OSH1*	LOC_Os03g51690	LH9	T15				24.25 ± 2.98	400	0	26	0	0	26
		RPY	T3	127.47 ± 31.73	80.64 ± 10.26	10.39 ± 2.76	26.16 ± 4.65	523.32 ± 96.86	0	9	261	6	276
*SP1*	LOC_Os11g12740	LH9	T2	128.1 ± 35.29	81.15 ± 9.51	9.56 ± 2.82	25.5 ± 2.92	545.93 ± 91.16	8	551	54	12	625
		RPY	T3	126.8 ± 31.39	80.47 ± 10.18	10.41 ± 2.76	26.01 ± 4.27	535.14 ± 99.06	2	6	224	1	233
*DTH8*	LOC_Os08g07740	LH9	T3	126.64 ± 31.35	80.40 ± 10.10	10.43 ± 2.76	25.97 ± 4.26	534.86 ± 98.71	2	231	40	4	277
		RPY	T1	127.79 ± 37.03	81.11 ± 9.98	9.49 ± 2.70	25.32 ± 3.48	521.64 ± 108.78	0	121	1471	40	1632
*DST*	LOC_Os03g57240	LH9	T2	127.97 ± 35.63	80.87 ± 9.94	9.55 ± 2.82	25.4 ± 3.21	533.27 ± 98.41	15	843	41	29	928
		RPY	T1	126.99 ± 35.56	81.26 ± 9.97	9.55 ± 2.70	25.4 ± 3.27	531.51 ± 102.87	7	58	1108	27	1200
*PYL4*	LOC_Os03g18600	LH9	T6	135.87 ± 56.44	78.08 ± 7.17		23.58 ± 3.67	337.22 ± 89.13	8	194	9	1	212
		RPY	T1	126.58 ± 35.19	81.25 ± 9.94	9.57 +2.72	25.44 ± 3.17	533.95 ± 100.24	0	92	1443	28	1563
*GNP1*	LOC_Os03g63970	LH9	T3	126.84 ± 31.59	80.46 ± 10.02	10.43 ± 2.77	25.86 ± 4.25	526.83 ± 105.58	8	581	36	22	647
		RPY	T1	126.56 ± 35.41	81.21 ± 9.88	9.52 ± 2.74	25.44 ± 3.27	535.59 ± 101.01	130	618	1941	150	2839

GNPP—grains number per panicle, FRET—fertility, EPN—effective panicle number, TGW—thousand grain weight, GYPM—grain yield per Mu.

**Table 4 ijms-24-01653-t004:** Genotypes of over 200 GNPP rice varieties in MBK database.

Name	Group	Origin	LAX1 LOC_Os01g61480	GNP1 LOC_Os03g63970
ZhongHan502	Japonica	China	T2	T6
Bg90-2	Intermediate(hybrid)	Sri Lanka	T6	T1
NingGeng28Hao	Japonica	China	T2	T1
YanGeng7Hao	Japonica	China	T5	T1
XiangQing	Japonica	China	T2	T1
C9083	Japonica	China	T2	T1
FUNAKIOMACHI	Japonica	Japan	T2	T1
HOUMANSHINDENINE	Japonica	Japan	T2	T1
KABASHIKO	Japonica	Japan	T2	T1
KAMEJI	Japonica	Japan	T2	T1
KAMENOO	Japonica	Japan	T2	T1
NingGeng24Hao	Japonica	China	T2	T1
RAIDEN	Japonica	Japan	T2	T1
WATARIBUNE1681	Japonica	Japan	T2	T1
CP231	Japonica	United States	T2	T1
Basmati370	Indica	India	T1	T5
Zhongchao 123	Japonica	China	T2	T21
ChangShu-6-85	Japonica	China	T2	T1
LianGeng11Hao	Japonica	China	T2	T1
PuTe6Hao	Japonica	China	T2	T1
SongGeng15	Japonica	China	T2	T1
TASENSHO	Japonica	Japan	T2	T1
R162	Japonica	China	T2	T1
NingGeng35Hao	Japonica	China	T13	T1
SHINYAMADABO1	Japonica	Japan	T2	T1
GORIKI	Japonica	Japan	T2	T1
MANGOKU	Japonica	Japan	T2	T1
JC1	Indica	India	T3	T1
SEKIYAMA	Japonica	Japan	T2	T1
AMBARIKORI	Indica	Africa	T1	T3

## Data Availability

Not applicable.

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
