# Peer review of "The Pyramiding of Elite Allelic Genes Related to Grain Number Increases Grain Number per Panicle Using the Recombinant Lines Derived from Indica–japonica Cross in Rice"

_ijms, 2023, doi:10.3390/ijms24021653_

Round 1

Reviewer 1 Report

The manuscript is focused on the genotype of known genes in rice breeding. The manuscript falls within the scope and contains useful information for the readers of IJMS. The idea of this manuscript is interesting. The analysis is good, and scientific contents are also good. However, the manuscript was not proof read before submission and it contains few syntax and grammar errors. The following are the specific comments and suggestions:

Point 1: Gene names format should be same in whole manuscript, better to use as italic.

Point 2: The full name can be replaced by an abbreviation in the Abstract L8.

Point 3: All scientific names should be italic such as Oryza sativa, indica, japonica etc. Please change in whole manuscript.

Point 4: Fig. 4: Figures should be closely relate to the content.

Point 5: In reference: Please correct the numbering.

Point 6: In reference: Please check that all the scientific names will be wrote in italics.

Point 7: L118-122, Need to re-phrase.

Point 8: Table 4: The authors concluded that LAX1-T2 could increase the grains number per panclie. Is there a reason to consider that there are more japonica in high grain number rice varieties.

Point 9: Fig. 2: Principal component analysis divided the group of RILs into two and how they were associated with the low, medium and high GNPP group.

Point 10: Table 2: the scores of each genotype whether need to be evaluated for significance to ensure the accuracy.

Point 11: Please confirm the selected parental material is representative of indica/japonica rice.

Author Response

Dear reviewer,

Our responses to your comments are highlighted in red type.

We thank the reviewer for appreciating of our work and for these comments, and the manuscript has been revised accordingly.

Reviewer 2 Report

・Lines 182 to 215 and Table 3 and Figure 4, which are included in "Discussion", should be described in "Results". Regarding Table 4, it is considered that there is no problem as it is as described in "Discussion".

Author Response

Dear reviewer,

Our responses to your comments are highlighted in red type. 

We thank the reviewer for the comment, the manuscript has been accordingly corrected as suggested.
